# Canonical Wnt signaling and the regulation of divergent mesenchymal Fgf8 expression in axolotl limb development and regeneration

**Giacomo L Glotzer[1†‡], Pietro Tardivo[1,2\*†], Elly M Tanaka[1\*]**

[1]Research Institute of Molecular Pathology (IMP), Vienna BioCenter (VBC), Campus-Vienna-Biocenter 1, Vienna, Austria; [2]Vienna BioCenter PhD Program, Doctoral School of the University of Vienna and Medical University of Vienna, Vienna, Austria

**\*For correspondence:**
pietro.tardivo@imp.ac.at (PT);
elly.tanaka@imp.ac.at (EMT)

[†]These authors contributed equally to this work

**Present address:** [‡]Yale University, New Haven, CT, United States

**Competing interest:** The authors declare that no competing interests exist.

**Abstract** The expression of fibroblast growth factors (Fgf) ligands in a specialized epithelial compartment, the Apical Ectodermal Ridge (AER), is a conserved feature of limb development across vertebrate species. In vertebrates, *Fgf 4*, *8*, *9*, and *17* are all expressed in the AER. An exception to this paradigm is the salamander (axolotl) developing and regenerating limb, where key Fgf ligands are expressed in the mesenchyme. The mesenchymal expression of Amex.*Fgf8* in axolotl has been suggested to be critical for regeneration. To date, there is little knowledge regarding what controls Amex.*Fgf8* expression in the axolotl limb mesenchyme. A large body of mouse and chick studies have defined a set of transcription factors and canonical Wnt signaling as the main regulators of epidermal *Fgf8* expression in these organisms. In this study, we address the hypothesis that alterations to one or more of these components during evolution has resulted in mesenchymal Amex.*Fgf8* expression in the axolotl. To sensitively quantify gene expression with spatial precision, we combined optical clearing of whole-mount axolotl limb tissue with single molecule fluorescent in situ hybridization and a semiautomated quantification pipeline. Several candidate upstream components were found expressed in the axolotl ectoderm, indicating that they are not direct regulators of Amex.*Fgf8* expression. We found that Amex.*Wnt3a* is expressed in axolotl limb epidermis, similar to chicken and mouse. However, unlike in amniotes, Wnt target genes are activated preferentially in limb mesenchyme rather than in epidermis. Inhibition and activation of Wnt signaling results in downregulation and upregulation of mesenchymal Amex.*Fgf8* expression, respectively. These results implicate a shift in tissue responsiveness to canonical Wnt signaling from epidermis to mesenchyme as one step contributing to the unique mesenchymal Amex.*Fgf8* expression seen in the axolotl.

## Editor's evaluation

The authors investigate the upstream regulators of axolotl Fgf8 expression in limb development and regeneration with the aim of identifying the reasons for its unique mesenchymal localization. The results suggest some refractoriness of the axolotl epidermis to activate Wnt signaling although Fgf8 expression in the anterior mesenchyme is still under Wnt control.

## Introduction

Signaling mediated by members of the fibroblast growth factor (Fgf) family is necessary for normal limb development and for limb regeneration in the salamander *Ambystoma mexicanum* (axolotl), a primary model for regeneration research (*Nacu et al., 2016*; *Purushothaman et al., 2019*). In

less-regenerative tetrapods, like mouse (*Mus musculus*), Fgf signaling is similarly required for limb development (*Lewandoski et al., 2000*; *Moon and Capecchi, 2000*; *Sun et al., 2002*; *Mariani et al., 2008*). In mammals, *Fgfs 4*, *8*, *9*, and *17* are produced by cells in the Apical Ectodermal Ridge (AER), a thickened epidermal structure at the distal dorsoventral boundary of the limb bud. Fgf signals from the AER act on the underlying mesenchyme to sustain its growth and patterning. Of the AER Fgfs, Mmu.*Fgf8* is the only one that is individually required for normal limb development in mice (*Mariani et al., 2008*).

Limb regeneration in the axolotl proceeds through the formation of the blastema, a structure that resembles the limb bud, both at the morphological and transcriptional level (*Gerber et al., 2018*). In both the axolotl limb blastema and limb bud, Amex.*Fgfs 8*, *9*, and *17* are expressed in the mesenchymal compartment, unlike in other vertebrates, while Amex.*Fgf4* transcription is negligible or absent (*Christensen et al., 2002*; *Nacu et al., 2016*; *Purushothaman et al., 2019*). Restriction of Amex.*Fgf8* expression to the mesenchymal compartment is to our knowledge unique to the salamander limb, where it serves a crucial role in connecting positional identity with growth and patterning during limb regeneration. Nacu et al. showed that Amex.*Fgf8* expression is restricted to the anterior mesenchyme of the regenerating blastema, while cells of the posterior blastema mesenchyme do not express Amex.*Fgf8* (*Nacu et al., 2016*). When a limb amputation is performed, anterior cells express Amex.*Fgf8* and posterior cells express Amex.*Shh*, initiating a positive feedback loop in which Amex.*Fgf8* and Amex.*Shh* reciprocally maintain each other's expression: prolonged expression of both signaling molecules sustains blastema growth and limb regeneration. The anterior mesenchymal expression of Amex.*Fgf8* therefore mediates the communication between anterior and posterior cells in the blastema. The incompetence of other blastema cells to transcribe Amex.*Fgf8* acts as a safeguard to restrict the initiation of blastema formation to injuries involving both anterior and posterior limb cells. It is interesting to note that the positive feedback relationship between *Fgfs* and *Shh* was initially described in the mammalian limb, between *Shh* in the posterior mesenchyme and *Fgf8* in the epidermis (*Laufer et al., 1994*; *Niswander et al., 1994*). Expression of Amex.*Fgf8* in the mesenchyme could serve to couple limb outgrowth with anterior–posterior positional information, a capacity that is restricted to mesenchymal cells in the axolotl (*Carlson, 1975*).

Given the interesting role mesenchymal Amex.*Fgf8* expression plays in axolotl limb regeneration, we asked: how has the axolotl limb signaling network been rewired to elicit this adaptation? In the mouse limb bud AER, expression of Mmu.*Fgf8* is controlled at the genetic level by a set of enhancers proximal to the gene locus (*Marinić et al., 2013*). Schlossnig et al. recently determined that conserved orthologs of the majority of these enhancers are present in the axolotl genome, and that at least one of these drives expression of a reporter construct in the mesenchyme of the axolotl limb (*Schloissnig et al., 2021*). This suggests that changes at the level of transacting factors may determine the divergent expression of Amex.*Fgf8* in the axolotl limb.

Transcription factors and intercellular signaling molecules that regulate AER-associated *Fgf8* expression have been identified in chicken (*Gallus gallus*) and mouse. In mice, AER expression of Mmu.*Fgf8* has been shown to be dependent on mesenchymal Mmu.*Fgf10* acting through the epidermis-specific receptor isoform Mmu.*Fgfr2b* (*Xu et al., 1998*; *De Moerlooze et al., 2000*; *Zhang et al., 2006*; *Sekine et al., 2019*). Using the chick model, it was established that Gga.*Fgf10* induces epidermal expression of the canonical Wnt ligand Gga.*Wnt3a*. *Wnt3a* (chick) and *Wnt3* (mouse), together with BMPR1a-dependent signaling, activate canonical Wnt signaling within AER cells to induce *Fgf8* (*Kengaku et al., 1998*; *Kawakami et al., 2001*; *Barrow et al., 2003*). In mice, genetic removal of epidermal Mmu.*Wnt3* or the canonical Wnt signaling effector β-catenin is sufficient to abolish Mmu.*Fgf8* expression in the limb, while expression of a constitutively active β-catenin markedly expands the Mmu.*Fgf8* expression domain and induces ectopic Mmu.*Fgf8* expression in the dorsal and ventral epidermis (*Barrow et al., 2003*; *Soshnikova et al., 2003*). Furthermore, mesenchymal expression of a constitutively active β-catenin construct appeared to potentially induce sporadic mesenchymal expression of Mmu.*Fgf8* (*Hill et al., 2006*). Similarly in the chick developing limb, misexpression of β-catenin, Gga.*Wnt3a* or the Wnt transcription factor Gga.*Lef1* is sufficient to induce Gga.*Fgf8* expression ectopically in the dorsal and ventral epidermis (*Kengaku et al., 1998*). In addition to Wnt signaling components, Buttonhead transcription factors *Sp8* and *Sp6* have also been shown to be necessary for *Fgf8* expression and act downstream of canonical Wnt signaling in the mammalian limb (*Kawakami et al., 2004*; *Talamillo et al., 2010*; *Lin et al., 2013*; *Haro et al., 2014*). Finally, Dlx factors *Dlx5* and

*Dlx6* have been implicated in the maintenance of the AER and are required to sustain *Fgf8* expression, but are dispensable for its induction (*Robledo et al., 2002*).

Little is known about the Amex.*Fgf8* regulatory machinery in the axolotl limb. Amex.*Fgf10* is expressed in the mesenchyme, along with *Fgfs 8*, *9*, and *17* (*Christensen et al., 2002*; *Nacu et al., 2016*; *Purushothaman et al., 2019*). This challenges the epithelial–mesenchymal interaction model of the mammalian and avian limb where mesenchymal *Fgf10* and epidermal *Fgf8* reciprocally sustain each other's expression acting on different, tissue-specific, receptor isoforms. A recent publication reported that no Fgf receptor is expressed in the axolotl epidermis (*Purushothaman et al., 2019*), suggesting that the epidermal side of the interaction may be completely absent. This is at odds with previous studies that showed *Fgfr2b* expression in the basal epidermis of the newt blastema (*Poulin and Chiu, 1995*). The expression domain of Amex.*Wnt3* ligands, the major *Fgf8* AER regulators, has not been determined in the axolotl limb although Wnt3a expression is present in the salamander limb blastema as determined by qPCR (*Singh et al., 2012*). Another canonical Wnt ligand, Amex.*Wnt10* was found in similar amounts in lateral injuries (surface wounds that do not trigger regeneration) and limb amputations, suggesting its role may be restricted to wound response (*Knapp et al., 2013*). Multiple studies reported that chemical activation of the canonical Wnt pathway increases cell proliferation in the salamander regenerating blastema (*Singh et al., 2012*; *Wischin et al., 2017*) but it has not been elucidated if the main site of canonical Wnt signaling activity is the blastema epidermis or mesenchyme. Although overexpression of the Wnt antagonist *Axin1* is sufficient to block limb development and regeneration in the axolotl (*Kawakami et al., 2006*), it is unclear if this effect is mediated through downregulation of Amex.*Fgf8*. In the axolotl limb, it has also not been determined whether the transcription factors Amex.*Sp6*, Amex.*Sp8*, Amex.*Dlx5*, Amex.*Dlx6*, and Amex.*Lef1* display mesenchymal or epidermal expression, nor if their function in regulating Amex.*Fgf8* is conserved.

Despite the expression of Fgf ligands in the mesenchymal compartment, classical experiments have shown that the epidermis of the axolotl blastema is necessary for both initial accumulation and proliferation of cells in the blastema (*Thornton, 1957*; *Campbell and Crews, 2008*). For example, Axolotl MARCKS like protein has been identified as an epidermal factor that induces the initial proliferation response in the blastema (*Sugiura et al., 2016*). Recent single-cell studies have highlighted that different cell populations are present in the regenerating axolotl epidermis (*Leigh et al., 2018*; *Rodgers et al., 2020*; *Li et al., 2021*), and previous studies have indicated that the basal layer of the axolotl epidermis expresses some markers of the mammalian AER, including Amex.*Fibronectin1* and Amex.*Dlx3* (*Christensen and Tassava, 2000*).

In this study, we take advantage of published single-cell RNA sequencing (scRNA seq) datasets for the axolotl limb bud (*Lin et al., 2021*) and blastema (*Li et al., 2021*) to elucidate the tissue compartment specificity of candidate Amex.*Fgf8* regulators. To precisely visualize tissue transcripts in the anatomical complexity of the developing and regenerating axolotl limb, we further develop a protocol that combines Hybridization Chain Reaction (HCR) in situ hybridization (*Choi et al., 2018*) with Ce3D tissue clearing (*Li et al., 2017*; *Li et al., 2019*) and light sheet imaging. We then implemented semiautomated image analysis pipelines to quantitatively compare transcript abundances in different tissue compartments and upon pharmacological perturbation.

We identify conserved and divergent features of the axolotl limb signaling network and implicate enhanced canonical Wnt signaling in the axolotl limb mesenchyme as one of the regulators of Amex.*Fgf8* mesenchymal expression.

## Results

### Expression of transcription factors known to regulate *Fgf8* in the chick and mouse AER

Several transcription factors have been functionally linked with the induction or maintenance of *Fgf8* in the mammalian AER (*Zuniga and Zeller, 2020*). To determine whether these transcription factors are potentially upstream of axolotl Amex.*Fgf8*, we investigated if they are expressed in the epidermis and/or mesenchyme in the axolotl developing and regenerating limb. To this end, we first analyzed published scRNA seq datasets for the axolotl stage 52 axolotl hindlimb bud (*Lin et al., 2021*) and 7-day postamputation axolotl forelimb blastema (*Li et al., 2021*). Both single-cell datasets were reanalyzed to match the latest axolotl transcriptome annotation (*Schloissnig et al., 2021*) and reclustered,

using known markers for cluster assignment (*Figure 1—figure supplements 1 and 2*). We then used HCR in situ hybridization in tissue sections and whole-mount cleared limb preparations of forelimb buds and blastemas of comparable stages (stage 45–46 forelimb bud, 7-day postamputation forelimb blastema) to validate the expression domains and acquire spatial information. To achieve precise visualization of transcripts in the anatomical complexity of the entire axolotl developing and regenerating limb, we combined HCR in situ hybridization with Ce3D tissue clearing and light sheet imaging (see Materials and methods).

Buttonhead transcription factors Mmu.*Sp6* and Mmu.*Sp8* are expressed in the AER of the mouse developing limb, and their knockout abolishes Mmu.*Fgf8* expression (*Lin et al., 2013*; *Haro et al., 2014*). Analysis of scRNA seq data from the axolotl limb bud and blastema (*Li et al., 2021*; *Lin et al., 2021*) revealed expression of Amex.*Sp6* and Amex.*Sp8* in the epidermis and not in mesenchyme (*Figure 1A, A', B, B'*). We performed whole-mount HCR staining in regenerating axolotl limb blastemas and limb buds (*Figure 2B, B'*) and confirmed that Amex.*Sp6* is expressed in the epidermis of the axolotl limb bud and regenerating blastema. Interestingly, we detected Amex.*Sp6* also in the mesenchyme of the axolotl limb bud, in a domain proximal to Amex.*Fgf8* expression (*Figure 2—figure supplement 1*). This proximal mesenchymal expression of Amex.*Sp6* was absent in the blastemas that we analyzed and is unlikely to regulate the more distal Amex.*Fgf8* expression in the axolotl limb bud. Using HCR in situ hybridization, we then found that Amex.*Sp8* is exclusively expressed in the epidermis (*Figure 2C, C'*), with a strong enrichment in the basal layer (the inner layer facing the mesenchyme) in both the axolotl and limb bud and blastema. This indicates that Buttonhead factors are unlikely to directly regulate Amex.*Fgf8* expression in the mesenchyme and that the lack of Amex.*Fgf8* expression in the axolotl epidermis is not due to the absence of these factors.

Dlx factors Mmu.*Dlx5* and Mmu.*Dlx6* are also expressed in the mouse AER, where they function to maintain Mmu.*Fgf8* expression (*Robledo et al., 2002*). Single-cell analysis of the axolotl blastema and limb bud detected transcripts in both the epidermal and mesenchymal compartments (*Figure 1C, C', D, D'*). Whole-mount HCR staining revealed that expression of Amex.*Dlx5* is enriched in the distal mesenchyme and in the basal epidermis of the blastema (*Figure 2D*) and limb bud (*Figure 2D'*). Expression of Amex.*Dlx5* in the limb epidermis suggests that in the axolotl, similar to the mouse, Amex.*Dlx5* is not sufficient to induce Amex.*Fgf8* expression. The strong mesenchymal Amex.*Dlx5* expression distinguishes the axolotl limb from the mammalian one and could be consistent with a conserved role in maintaining Amex.*Fgf8* expression.

The canonical Wnt pathway is a main regulator of *Fgf8* expression in the chicken and mouse developing limb. In the chick, the Wnt transcription factor Gga.*Lef1* is required for AER Gga.*Fgf8* expression. Furthermore, Gga.*Lef1* overexpression is sufficient to induce ectopic Gga.*Fgf8* expression in the non-AER epidermis (*Kengaku et al., 1998*). In mouse, the double knockout of the Wnt transcription factors Mmu.*Lef1* and Mmu.*Tcf1* abolishes AER Mmu.*Fgf8* expression (*Galceran et al., 1999*). In our scRNA seq analysis, we could detect expression of Amex.*Lef1* in both epidermal and mesenchymal cells (*Figure 1E, E'*). Using HCR, we found that Amex.*Lef1* is expressed broadly in the limb bud and blastema mesenchyme (*Figure 2E, E'*). We also detected expression in the limb bud and blastema epidermis, strongest in the distal portion of the basal epidermis (*Figure 2E, E'*). Considering that overexpression of Gga.*Lef1* in chick is sufficient to induce ectopic Gga.*Fgf8* expression (*Kengaku et al., 1998*), we note here that, at least at the levels observed, it is not sufficient to allow Amex.*Fgf8* expression in the axolotl limb epidermis. The broad expression pattern of Amex.*Lef1* in the axolotl limb mesenchyme could be compatible with a role in inducing or maintaining mesenchymal Amex.*Fgf8* expression in axolotl.

Our analysis revealed that the transcription factors Amex.*Sp8*, Amex.*Sp6*, Amex.*Dlx5*, Amex.*Dlx6*, and Amex.*Lef1* are expressed in the axolotl developing and regenerating limb epidermis, in particular all are present in the basal epidermis. This indicates that, in the axolotl limb epidermis, the combined endogenous expression of these factors is not sufficient to activate Amex.*Fgf8* transcription. While expression of Sp factors appeared uniquely epidermal, Amex.*Dlx5*, Amex.*Dlx6*, and Amex.*Lef1* are expressed in the limb bud and blastema mesenchyme, leaving open the possibility that they have a role in the regulation of Amex.*Fgf8* expression in this tissue compartment.

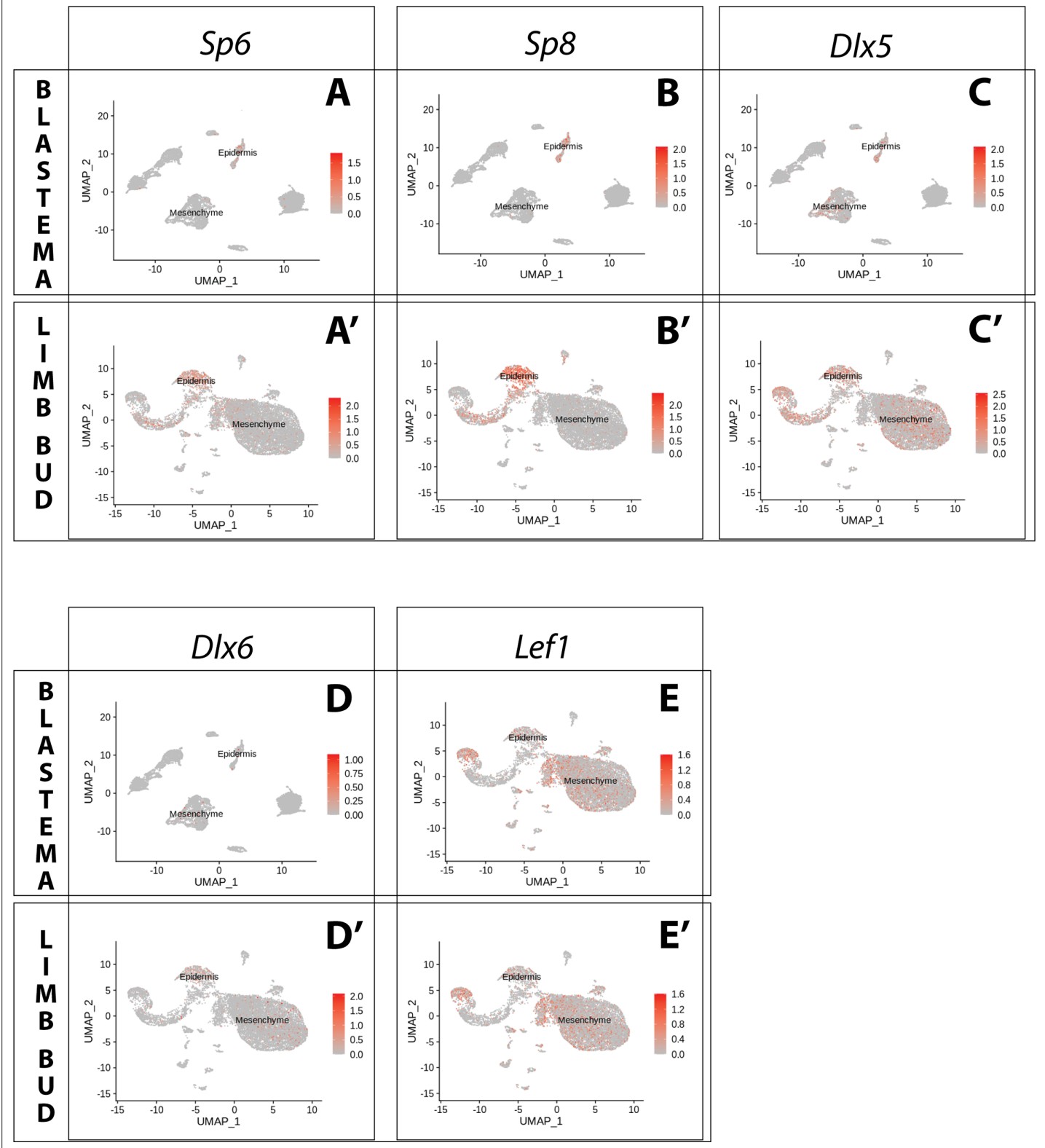

**Figure 1.** Expression of AER transcription factors in the axolotl limb bud and limb blastema assessed by reanalysis of Axolotl scRNA seq datasets. Reanalysis of scRNA seq data from blastema (**A–E**, data from *Li et al., 2021*) or limb bud (**A'–E'**, data from *Lin et al., 2021*). (**A, A'**) UMAPs of Axolotl scRNA seq expression data for *Sp6*. Expression is detected in the epidermis of the axolotl blastema and limb bud. (**B, B'**) UMAPs of Axolotl scRNA seq expression data for *Sp8*. Expression is detected in the epidermis of the axolotl blastema and limb bud. (**C, C'**) UMAPs of Axolotl scRNA seq expression

*Figure 1 continued on next page*

*Figure 1 continued*

data for *Dlx5*. Expression is detected in the epidermis and mesenchyme of the axolotl blastema and limb bud. (**D, D'**) UMAPs of Axolotl scRNA seq expression data for *Dlx6*. Expression is detected in the epidermis and mesenchyme of the axolotl limb bud. Expression is detected only in few blastema cells. (**E, E'**) UMAPs of Axolotl scRNA seq expression data for *Lef1*. Expression is detected in the epidermis and mesenchyme of the axolotl blastema and limb bud.

The online version of this article includes the following figure supplement(s) for figure 1:

**Figure supplement 1.** Cluster identification in the reanalysis of axolotl limb bud scRNA seq dataset from *Lin et al., 2021*.

**Figure supplement 2.** Cluster identification in the reanalysis of axolotl blastema scRNA seq dataset from *Li et al., 2021*.

## Conserved and divergent expression of canonical Wnt pathway components and upstream regulators

Canonical Wnt signaling controls AER-associated *Fgf8* expression and mediates the communication between the epidermal and mesenchymal limb compartments in chick and mouse. To verify if this interaction is conserved in axolotl, we decided to investigate the expression of the canonical Wnt3 ligands and of the components that are known to induce their expression in the mammalian limb bud AER, primarily *Fgf10* and its receptor *Fgfr2b*. We furthermore profiled *Rspo2*, an extracellular potentiator of Wnt signaling that is necessary for normal limb development (*Xu et al., 1998*; *De Moerlooze et al., 2000*; *Zhang et al., 2006*; *Sekine et al., 2019*). Finally, to localize the site of canonical Wnt signaling, we examined the expression of *Axin2*, a target of the canonical Wnt pathway that is often used as a readout of signaling activation strength (*Nam et al., 2007*; *Itou et al., 2011*; *Eckei et al., 2016*; *Nusse and Clevers, 2017*).

### Conserved gene expression features

In both blastema and limb bud scRNA seq datasets, we detected Amex.*Fgf10* transcripts in the mesenchymal cluster (*Figure 3A, A'*), consistent with previous reports (*Christensen et al., 2002*; *Nacu et al., 2016*; *Purushothaman et al., 2019*). Using HCR in situ hybridization we confirmed strong and broad mesenchymal expression (*Figure 4B, B'*). Surprisingly, we additionally found Amex.*Fgf10* transcripts in a small and distinct cell population located in the basal layer of the distal epidermis. To exclude that this population is of mesenchymal origin we performed HCR in situ hybridization and lineage tracing using *Prrx1*-Cre transgenic axolotls (*Figure 4—figure supplement 1*; *Gerber et al., 2018*). During mammalian limb development, mesenchymal *Fgf10* signals to the epidermis via the specific receptor isoform *Fgfr2b* (*Xu et al., 1998*; *De Moerlooze et al., 2000*; *Zhang et al., 2006*; *Sekine et al., 2019*). In the axolotl, using HCR, we detected Amex.*Fgfr2b* transcripts through the entire basal epidermis of the blastema (*Figure 4C*) and limb bud (*Figure 4C'*), consistent with Amex.*Fgf10* signaling to the epidermis as in other vertebrates.

   *Fgf10* signaling is thought to be required for expression of Mmu.*Wnt3* in mouse and Gga.*Wnt3a* in chick. We therefore assessed expression of both Amex.*Wnt3* and Amex.*Wnt3a*: we found Amex.*Wnt3a* transcripts in the epidermal cluster (*Figure 3B, B'*), while we could only detect negligible Amex.*Wnt3* transcripts in any cell type. HCR data confirmed that Wnt3a expression is present in the axolotl limb bud and blastema basal epidermis (*Figure 4D, D'*). These results suggest that the epidermal induction of Amex.*Wnt3a* is downstream of mesenchymal Amex.*Fgf10* signaling in the axolotl limb as in mouse.

### Wnt signaling activity and the Rspo2 coactivator show divergent gene expression features

In mouse and chick limb development, epidermal signaling from Mmu.*Wnt3* or Gga.*Wnt3a* is considered to induce canonical Wnt signaling in the AER (*Xu et al., 1998*; *De Moerlooze et al., 2000*; *Zhang et al., 2006*; *Sekine et al., 2019*). This signal activation results in strong AER expression of *Axin2*, a canonical Wnt target that is often used to determine pathway activity (*Nam et al., 2007*; *Itou et al., 2011*; *Eckei et al., 2016*; *Nusse and Clevers, 2017*). *Wnt3* ligands act also on the underlying mesenchyme, where they induce expression of *Axin2*, albeit at lower levels (*Kengaku et al., 1998*).

   To determine the prevalent site of canonical Wnt signaling in the axolotl limb, we evaluated the expression of Amex.*Axin2*. Single-cell analysis indicated that Amex.*Axin2* expression in the axolotl limb is biased toward the mesenchymal compartment (*Figure 3C, C'*). Using whole-mount HCR,

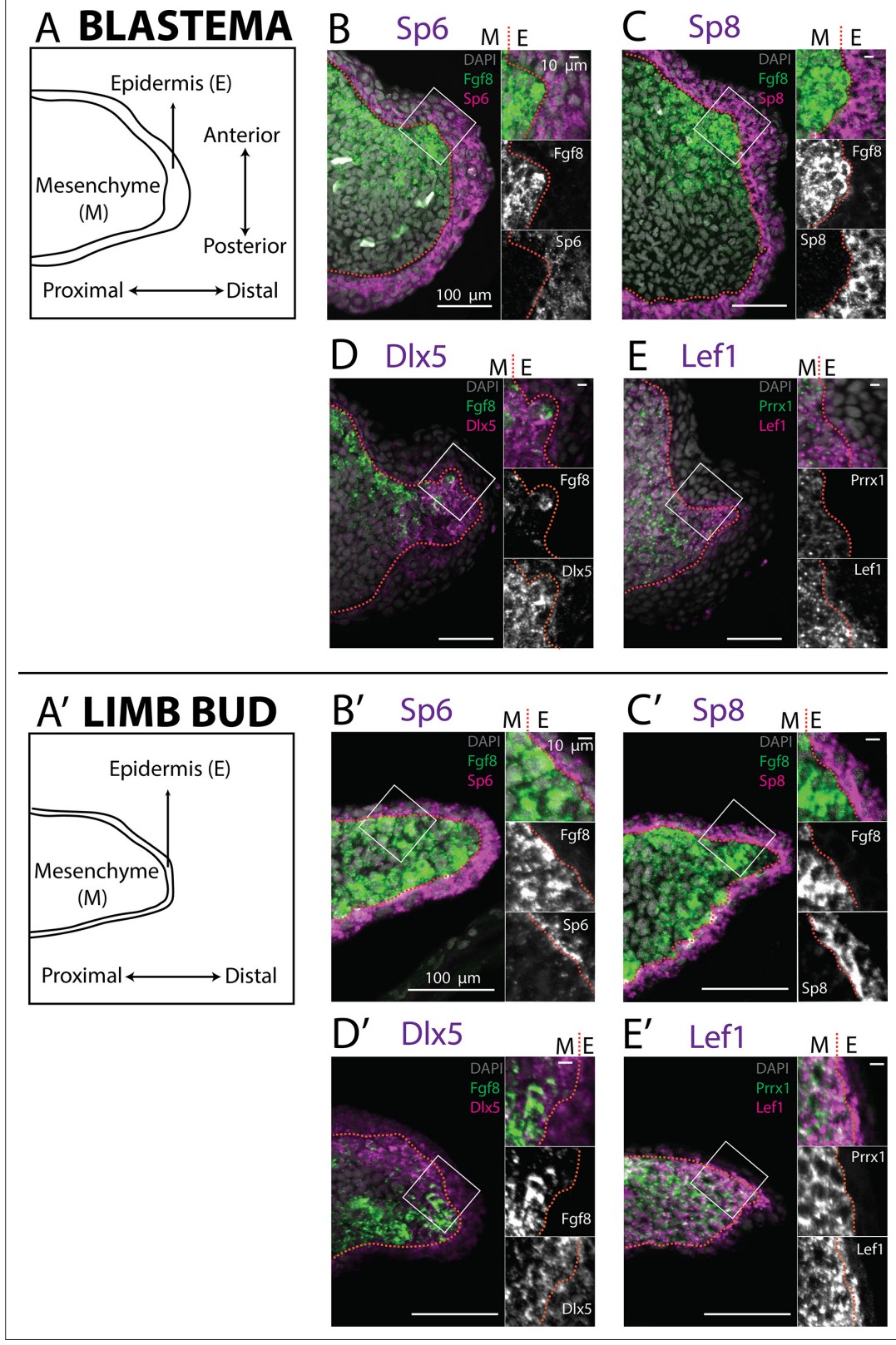

**Figure 2.** Expression of AER transcription factors in the axolotl limb blastema and limb bud assessed by HCR in situ hybridization. (**A, A′**) Schematic outlining the mesenchymal and epidermal compartments in a longitudinal section of an axolotl blastema and limb bud. (**B, B′**) Expression of *Sp6* in the epidermis and of *Fgf8* in the mesenchyme of the axolotl blastema and limb bud revealed by HCR (single planes from whole-mount images, *n* =

*Figure 2 continued on next page*

*Figure 2 continued*

4). (**C, C′**) Expression of *Sp8* in the epidermis and of *Fgf8* in the mesenchyme of the axolotl blastema and limb bud revealed by HCR (single planes from whole-mount images, *n* = 4). (**D, D′**) Expression of *Fgf8* in the mesenchyme as well as of *Dlx5* in the mesenchyme and basal epidermis of the axolotl limb bud and blastema (single planes from whole-mount images, *n* = 4). (**E, E′**) Expression of the mesenchymal marker *Prrx1*, and of *Lef1* in the mesenchyme and basal epidermis of the axolotl blastema and limb bud (single planes from whole-mount images, *n* = 4). For microscopy images right panels represent magnifications of the outlined boxes, M = mesenchyme, E = epidermis. Dashed lines demarcate epidermal–mesenchymal boundaries.

The online version of this article includes the following figure supplement(s) for figure 2:

**Figure supplement 1.** *Sp6* is expressed in the basal epidermis and proximal mesenchyme in the axolotl limb bud (*n* = 4).

---

we confirmed strong expression in the mesenchyme, with a distal and subepidermal enrichment in both the axolotl blastema (*Figure 4D, E*) and limb bud (*Figure 4D′, E′*). Epidermal expression is also present, but weaker. The mesenchymal enrichment in Amex.*Axin2* expression distinguishes the axolotl limb from the mammalian one.

Mmu.*Rspo2*, a secreted ligand that acts as an enhancer for Wnt signaling, is expressed in the mouse limb AER where it functions to potentiate canonical Wnt signaling activity. Mutations in the *Rspo2* gene cause severe limb truncations in mice, and complete loss of limbs in humans (*Nam et al., 2007*; *Aoki et al., 2008*; *Bell et al., 2008*; *Yamada et al., 2009*; *Szenker-Ravi et al., 2018*). Our single-cell analysis revealed that Amex.*Rspo2* expression is absent from the axolotl developing and regenerating limb epidermis but present in the mesenchyme (*Figure 3D, D′*). We confirmed the strictly mesenchymal expression of Amex.*Rspo2* using HCR (*Figure 4E, E′*). In both the axolotl limb bud and blastema, mesenchymal Amex.*Rspo2* expression is biased along the dorsoventral axis, suggesting a possible role in dorsoventral limb patterning (*Figure 4—figure supplement 2*).

## Amex.*Fgf8* expression is responsive to pharmacological perturbation of Wnt signaling

The mesenchymal enrichment of Amex.*Axin2* transcripts prompted us to test if canonical Wnt signaling regulates Amex.*Fgf8* expression in the mesenchyme of the axolotl regenerating limb, and whether enhancing Wnt signaling in the epidermis would be sufficient to induce ectopic Amex.*Fgf8* expression. To this end, we treated regenerating axolotls with a chemical Wnt agonist or antagonist and used HCR in whole-mount blastema samples to visualize the effect on transcription of Amex.*Fgf8*, using Amex.*Axin2* as a readout for successful pathway perturbation. To accurately evaluate the effect of the drug treatments, we developed a semiautomated image analysis pipeline that enables the segmentation of gene expression domains, the removal of the signal derived from autofluorescent blood cells in the blastema, and the quantitative comparison of gene expression levels (*Figure 5—figure supplement 1*, Materials and methods).

First, to assess the possible role of Wnt signaling in maintaining Amex.*Fgf8* expression, we performed drug treatments in regenerating blastemas. Axolotls were amputated at mid-zeugopod level, allowed to form a blastema over 5 days and then were bathed for 12 hr in a 25 µM solution of IWR-1-endo (Sigma), an inhibitor of the canonical Wnt pathway that acts by stabilizing Axin2 and consequently increasing the degradation of β-catenin. IWR treatment induced a similar decrease in Amex.*Axin2* and Amex.*Fgf8* transcripts when compared to Dimethyl sulfoxide (DMSO) treated controls (*Figure 5A, B*), suggesting that canonical Wnt signaling is required to sustain expression of Amex.*Fgf8* in the axolotl blastema mesenchyme.

Second, to assess whether canonical Wnt signaling can promote Amex.*Fgf8* expression, axolotl with 5-day blastemas were bathed in a solution of 50 µM CHIR-99021 (Tocris) or in DMSO control for 3, 6 and, in a separate experiment, 12 hr. As expected, Amex.*Axin2* expression was upregulated in all CHIR treatment conditions compared to controls (*Figure 5C*). After 3 hr of CHIR treatment, expression of Amex.*Fgf8* was significantly upregulated (*Figure 5D*). By 6 hr, we found that Amex.*Fgf8* was not significantly different in CHIR-treated and -untreated samples while after 12-hr Amex.*Fgf8* expression levels were significantly decreased when compared to control samples (*Figure 5D*).

Amex.*Fgf8* transcripts, although upregulated after 3 hr of CHIR treatment, were always present exclusively in the native Amex.*Fgf8* domain in the anterior blastema mesenchyme (*Figure 5—figure*

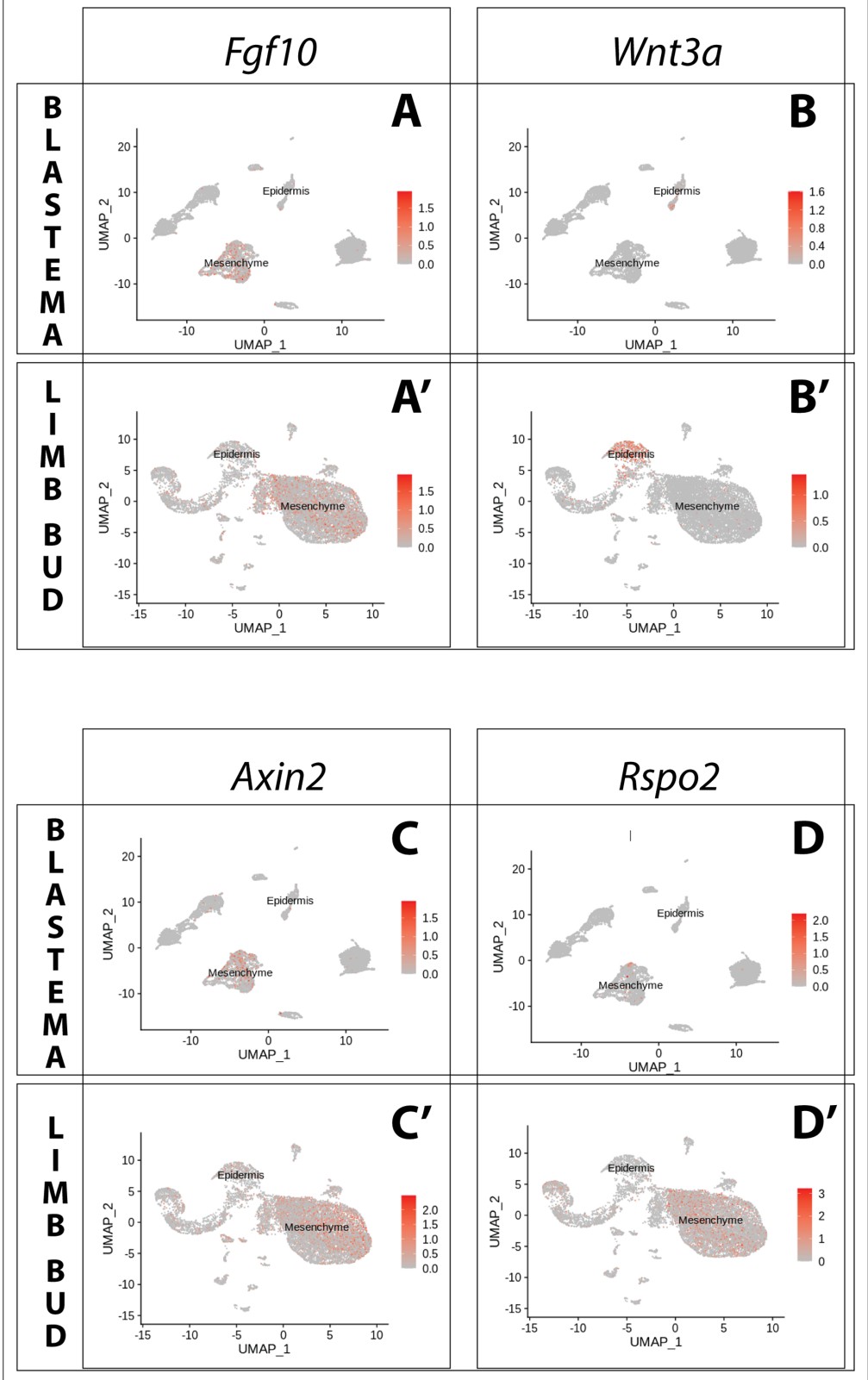

**Figure 3.** Expression of *Fgf10* and of the canonical Wnt pathway components *Wnt3a, Axin2, Rspo2* in the axolotl limb bud and limb blastema assessed by reanalysis of Axolotl scRNA seq datasets. Reanalysis of scRNA seq data from blastema (**A–D**, data from *Li et al., 2021*) or limb bud (**A'–D'**, data from *Lin et al., 2021*). (**A, A'**) UMAPs of Axolotl scRNA seq expression data for *Fgf10*. Expression is detected in the mesenchyme of the axolotl blastema

*Figure 3 continued on next page*

*Figure 3 continued*

and limb bud. (**B, B'**) UMAPs of Axolotl scRNA seq expression data for *Wnt3a*. Expression is detected in the epidermis of the axolotl blastema and limb bud. (**C, C'**) UMAPs of Axolotl scRNA seq expression data for *Axin2*. Expression is detected in the prevalently in the mesenchyme of the axolotl blastema and limb bud. (**D, D'**) UMAPs of Axolotl scRNA seq expression data for *Rspo2*. Expression is detected in the mesenchyme of the axolotl limb bud. Expression is detected only in few blastema cells.

*supplement 2*). Our results are consistent with canonical Wnt signaling regulating Amex.*Fgf8* expression in the anterior mesenchyme. Interestingly, while we observe a Wnt-induced upregulation of Amex.*Fgf8* in the anterior mesenchyme, Wnt activation is not sufficient to induce ectopic Amex.*Fgf8* expression in the posterior mesenchyme nor in the epidermis indicating regulation of Amex.*Fgf8* by other, yet unidentified factors. Surprisingly, we observed that in all treatment conditions Amex.*Axin2* expression retained a strong mesenchymal bias, suggesting that the axolotl epidermis may be partially refractory to canonical Wnt signaling activation. To verify this, we quantified the increase in Amex.*Axin2* in the mesenchymal and epidermal compartments after 3 and 6 hr of CHIR treatment and found that upregulation of Amex.*Axin2* is present in both compartments, but stronger in the mesenchyme (*Figure 5E*, *Figure 5—figure supplement 3*).

## Discussion

Salamanders are the sole modern tetrapod capable of regenerating their limbs throughout their lifespans. Along with a remarkable regenerative ability, the salamander limb presents other distinctive features in its development, both at the morphological and molecular level. The digital elements are formed in an anterior to posterior order, contrary to other species (*Nye et al., 2003*; *Fröbisch et al., 2015*; *Purushothaman et al., 2019*). Furthermore, key molecular players of limb development, such as Fgf ligands *Fgf 8*, *9*, and *17*, exhibit divergent expression in salamander limb development and regeneration when compared to mouse and chick (*Nacu et al., 2016*; *Purushothaman et al., 2019*).

We have focused on the unique mesenchymal expression of Amex.*Fgf8* with the aim of establishing how the upstream signaling circuit characterized in amniotes differs in axolotl. We found that most of the components that act upstream of mammalian *Fgf8* expression in the AER are present in the axolotl epidermis, with an enrichment in the most basal layer. These include Amex.*Fgfr2b*, the receptor for *Fgf10*, Amex.*Wnt3a*, and the transcription factors Amex.*Sp8*, Amex.*Sp6*, Amex.*Dlx5*, Amex.*Dlx6*, and Amex.*Lef1*. Expression of these components in the axolotl epidermis is evidently insufficient to induce expression of Amex.*Fgf8* in epidermis. This deficit could be explained by the acquisition of completely new and different regulators of Amex.*Fgf8* in the axolotl limb, or by the presence of components that repress Amex.*Fgf8* expression in the axolotl epidermis, or by the lack of a different necessary regulator that may be present in the mouse epidermis but absent in the axolotl epidermis. Epidermal expression of Amex.*Fgfr2b* and Amex.*Wnt3a* suggests that in the axolotl limb, like in the mouse and chick limb, Amex.*Fgf10* signals to the epidermis to induce the expression of a canonical *Wnt3* ligand. There are slight differences in the epidermal pattern of expression of *Wnt3* ligands between the chick and mouse limb: in chick Gga.*Wnt3a* expression is restricted to the AER (*Kawakami et al., 2001*), while in mouse Mmu.*Wnt3* expression is present in the entire limb epidermis (*Barrow et al., 2003*). Nevertheless, it has been speculated that in both species, Mmu.*Wnt3*/Gga.*Wnt3a* ligands signal to induce strong Wnt signaling activation specifically in the AER, a cell type that has an enhanced sensitivity to canonical Wnt signaling (*Kengaku et al., 1998*; *Barrow et al., 2003*). In the axolotl, unlike the mouse, we found that a discrete epidermal compartment exhibiting high activity of the canonical Wnt pathway is absent, while strong canonical Wnt pathway activity appears to be present in the mesenchyme. Our data suggest that the axolotl has lost competency to activate strong canonical Wnt signaling in the epidermis and diverted this competence to the mesenchyme. Interestingly, *Rspondin2*, a potentiator of Wnt signaling in the mouse epidermis is expressed strictly in the mesenchyme in axolotl.

Pharmacological activation and inhibition of the Wnt pathway yielded results consistent with a role of Wnt signaling in promoting mesenchymal Amex.*Fgf8* expression. A positive effect of CHIR treatment on Amex.*Fgf8* expression was present after 3 hr of treatment but disappeared when the perturbation was performed for longer durations, possibly due to indirect effects or negative feedbacks

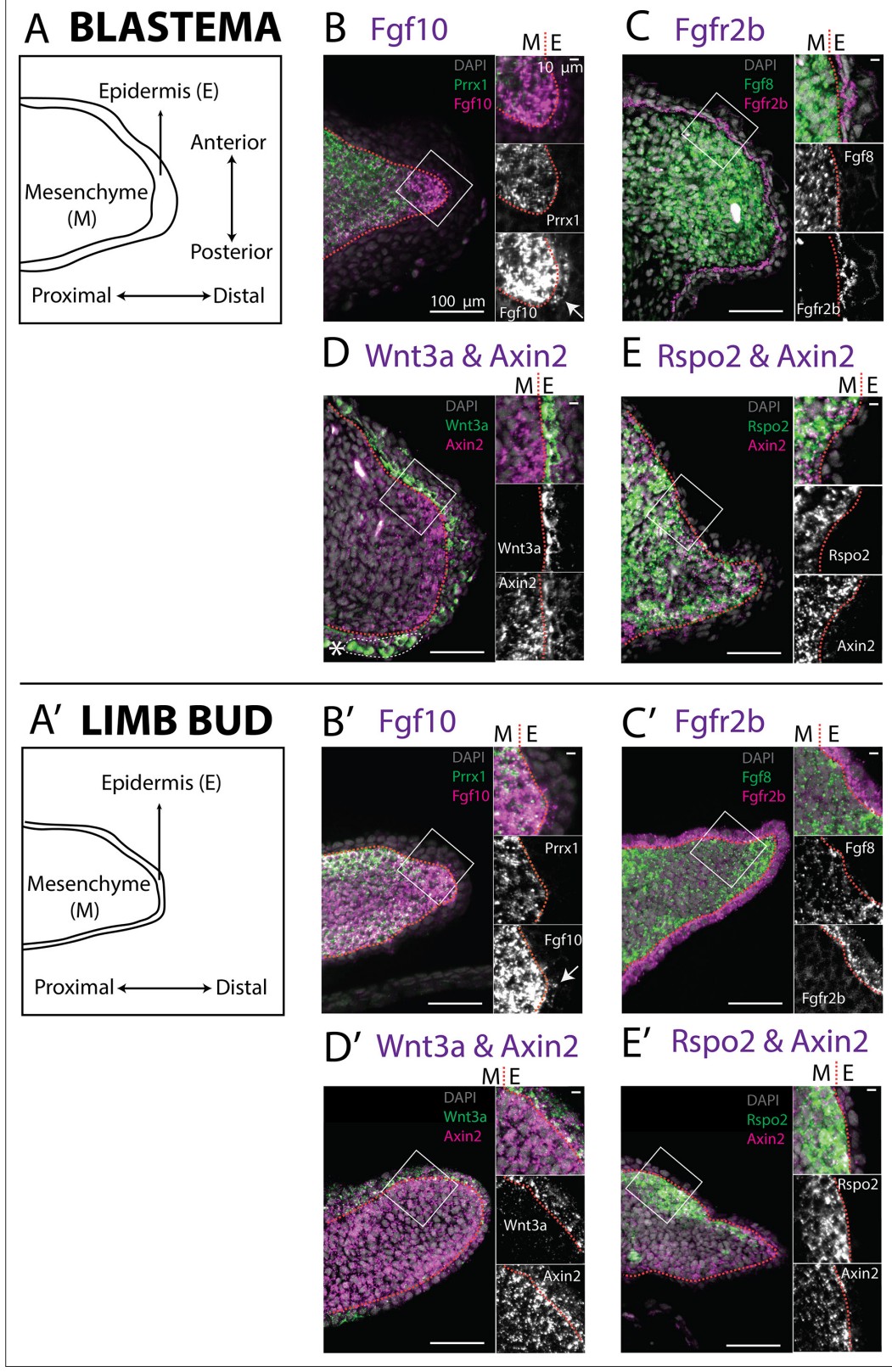

**Figure 4.** Expression of *Fgf10*, *Fgfr2b* and of the canonical Wnt pathway components *Wnt3a*, *Axin2*, *Rspo2* in the axolotl limb bud and limb blastema evaluated by HCR in situ hybridization. (**A, A'**) Schematic outlining the mesenchymal and epidermal compartments in a longitudinal section of an axolotl blastema and limb bud. (**B, B'**) Expression of *Prrx1* and *Fgf10* in the mesenchyme of the axolotl blastema and limb bud revealed by HCR

*Figure 4 continued on next page*

*Figure 4 continued*

(single planes from whole-mount images, *n* = 4). Arrows point to weak *Fgf10* expression present in the distal basal epidermis of the blastema and limb bud. (**C, C'**) Expression of *Fgf8* in the mesenchyme and expression of *Fgfr2b* in the basal epidermis of the axolotl blastema and limb bud revealed by HCR (single planes from whole-mount images, *n* = 4). (**D, D'**) Expression of *Wnt3a* in the basal epidermis and *Axin2* in the mesenchyme and basal epidermis of the axolotl blastema and limb bud (single planes from whole-mount images, *n* = 4). *Axin2* expression is stronger in the mesenchyme and weaker in the epidermis. (**E, E'**) Expression of *Rspo2* in the mesenchyme and *Axin2* in the mesenchyme and basal epidermis of the axolotl blastema and limb bud (single plane from whole-mount image, *n* = 4). Bright green structures (*) in the blastema outer epidermis are autofluorescent signal. For microscopy images right panels represent magnifications of the outlined boxes, M = mesenchyme, E = epidermis. Dashed lines demarcate epidermal–mesenchymal boundaries.

The online version of this article includes the following figure supplement(s) for figure 4:

**Figure supplement 1.** Epidermal cells expressing *Fgf10* are not of mesenchymal origin.

**Figure supplement 2.** *Rspo2* expression is biased along the dorsoventral axis in the axolotl blastema (**A–C**) and limb bud (**D–F**) (*n* = 4).

---

triggered by the ubiquitously high Wnt signaling environment. The positive upstream regulatory role of Wnt signaling is similar to that seen in chicken and mammals yet occurring in a different tissue compartment (***Barrow et al., 2003***; ***Soshnikova et al., 2003***). Our experiments have been performed at stages of limb development and regeneration subsequent to the initiation of Amex.*Fgf8* expression and therefore indicate a function of canonical Wnt signaling at least in maintaining Amex.*Fgf8* expression. In mouse and chick, canonical Wnt signaling is required for both the initiation and maintenance of endogenous Fgf8 expression and, when activated ectopically, can induce ectopic epidermal Fgf8 expression (***Kengaku et al., 1998***; ***Galceran et al., 1999***; ***Barrow et al., 2003***; ***Soshnikova et al., 2003***; ***Kawakami et al., 2004***). Interestingly, in the axolotl, canonical Wnt activation was not sufficient to induce ectopic Amex.*Fgf8* expression in the ectoderm. This difference may depend on a regulative process that is species specific or on the method of experimental perturbation. Classical mouse and chick studies have not fully clarified if the Wnt-dependent induction of ectopic *Fgf8* expression is a direct and cell autonomous phenomenon. Ectopic *Fgf8*-expressing cells were often generated in stripes, suggesting that Wnt signaling may act to induce a local tissue reorganization that is likely dependent on additional signaling components, leaving open the possibility that β-catenin pathway activation may not be sufficient to induce *Fgf8* expression cell autonomously in the mammalian and avian limb (***Kengaku et al., 1998***; ***Barrow et al., 2003***; ***Soshnikova et al., 2003***). Of the transcription factors that have been implicated in the regulation of AER *Fgf8* expression, Amex.*Lef1*, Amex.*Dlx5*, and Amex.*Dlx6* were expressed also in the axolotl mesenchyme, where they may play a role in inducing or sustaining Amex.*Fgf8* expression. Nevertheless, their broad expression pattern includes portions of the limb mesenchyme devoid of Amex.*Fgf8* transcripts, suggesting they are not sufficient to induce Amex.*Fgf8* expression outside of its native domain.

In our study, we also report that Amex.*Fgf10* expression is present not only in the axolotl limb mesenchyme but also in a small cell population localized in the basal layer of the limb bud and blastema epidermis. This additional expression domain is to our knowledge unique to the axolotl; *Fgf10* is considered a mesenchymal specific ligand, and the mesenchymal to epidermal *Fgf10–Fgfr2b* signaling interaction is conserved in the development of different organs across species, likely including the axolotl limb (***Itoh, 2016***) (and this study). We show that the additional Amex.*Fgf10*-expressing population in the axolotl limb is not derived from the blastema mesenchyme and does not express the mesenchymal marker Amex.*Prrx1*. Consequently, it would be of interest to assess the function of these cells and of Amex.*Fgf10* expression in this additional domain, as well as to determine their lineage origin. They could be of ectodermal origin or derive from a different mesodermal linage. Interestingly, Masselink et al. showed that mesodermal cells of somitic origin are present in the zebrafish pectoral fin AER and that ablation of this cell population expands the Dre.*Fgf8* expression domain in the fin ectoderm (***Masselink et al., 2016***). Loss of this somitic mesoderm-derived cell population in the tetrapod limb has been speculated to be necessary for temporal expansion of *Fgf8* AER expression and fin to limb transition (***Thorogood, 1992***). It is possible that in axolotl, where Amex.*Fgf8* expression is not epidermal, a somitic population has been maintained in the epidermis and carries functions that could be important for regeneration.

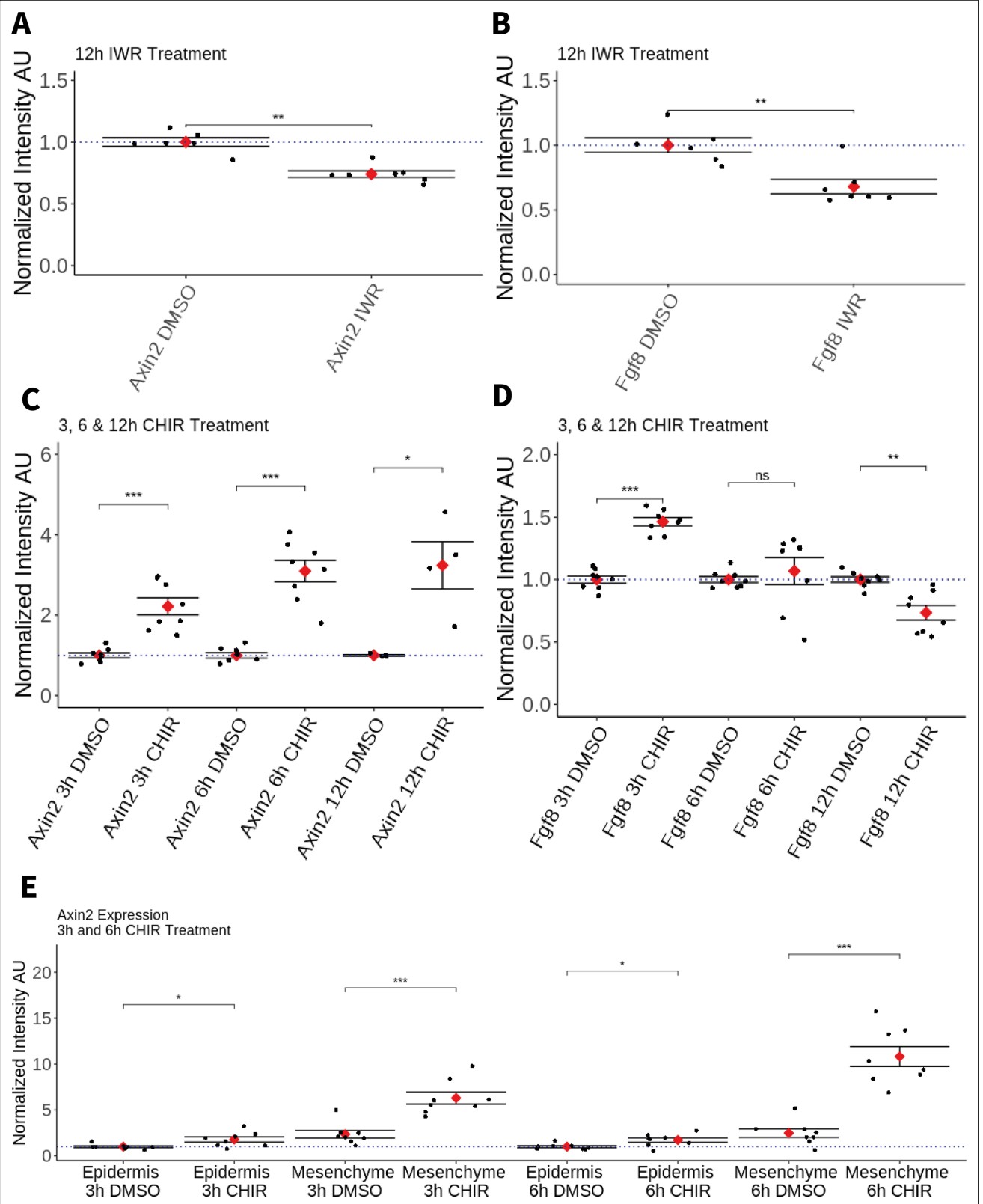

**Figure 5.** Effect of pharmacological perturbation of Wnt signaling on *Fgf8* expression in the axolotl blastema. Plots depicting expression levels of *Fgf8* or *Axin2* as assessed by mean fluorescent intensity of HCR signal inside the corresponding gene expression domains. Each black dot represents the mean signal quantified from one blastema, each red dot represents mean values for each condition. Whiskers indicate the standard error of the mean. * indicates statistical significance (*p < 0.05, **p < 0.01, ***p < 0.001, ns = nonsignificant = p > 05). Statistics were performed using the nonparametric

*Figure 5 continued on next page*

*Figure 5 continued*

Wilcoxon rank sum test. (**A, B**) Pharmacological inhibition of Wnt signaling using IWR1-endo downregulates *Axin2* and *Fgf8* expression in the axolotl limb blastema. (**C**) CHIR treatment activates canonical Wnt signaling in the axolotl blastema as shown by upregulation of *Axin2* expression after 3, 6, and 12 hr of treatment. (**D**) Pharmacological activation of Wnt signaling using CHIR upregulates *Fgf8* in the axolotl blastema after 3 hr of treatment. *Fgf8* expression comparable to DMSO control levels after 6 hr of treatment and downregulated after 12 hr of CHIR treatment. (**E**) Mesenchymal and epidermal *Axin2* expression are both upregulated after 3 and 6 hr of CHIR treatment. Mesenchymal *Axin2* increases 2.6-fold after 3 hr and 4.4-fold after 6 hr of treatment. Epidermal *Axin2* increases 1.8-fold after 3 hr and 1.7-fold after 6 hrs of treatment.

The online version of this article includes the following source data and figure supplement(s) for figure 5:

**Source data 1.** Numerical data for the plots in *Figure 5*.

**Figure supplement 1.** Image analysis workflow for whole-mount HCR signal quantification inside gene expression domains.

**Figure supplement 2.** Single plane images from whole-mount stacks of 3-hr CHIR-treated and DMSO control samples.

**Figure supplement 3.** Image analysis workflow for the quantification of *Axin2* HCR signal quantification in the epidermal and mesenchymal domains.

Further studies of axolotl limb patterning will help refine our understanding of the unique mechanisms that salamanders evolved to make a limb. Some of the adaptations of the axolotl limb patterning system may be important to accommodate specific requirements of limb regeneration, while features that are conserved across species may represent necessary elements of limb development. Mesenchymal expression of Amex.*Fgf8* is of particular interest as it has been implicated in anterior–posterior patterning and in the intercalation capability in limb regeneration (*Shimizu-Nishikawa et al., 2003*; *Makanae and Satoh, 2018*). Our study highlights that known regulators of AER *Fgf8* expression are present in the axolotl epidermis, where they are not sufficient to induce Amex.*Fgf8* expression and suggests that an epidermal to mesenchymal trade in sensitivity to canonical Wnt signaling may be an important factor for mesenchymal expression of Amex.*Fgf8* and a unique feature of limb development and regeneration in the axolotl or salamanders in general.

## Materials and methods

### Key resources table

| Reagent type (species) or resource | Designation | Source or reference | Identifiers | Additional information |
|---|---|---|---|---|
| Strain, strain background (*Ambystoma mexicanum, d/d* strain) | Axolotl, d/d strain | Tanaka lab axolotl colony | – | Axolotl stock maintained in Tanaka lab, Vienna, Austria. |
| Chemical compound, drug | CHIR-99021 | Tocris | 4423 | |
| Chemical compound, drug | IWR-1, ≥98% (hplc) | Sigma-Aldrich | I0161-25MG | |
| Chemical compound, drug | DAPI | Sigma-Aldrich | D9542 | |
| Chemical compound, drug | DMSO | Sigma-Aldrich | D2650-5 × 5 ML | |
| Chemical compound, drug | Histodenz | Sigma-Aldrich | D2158-100G | |
| Chemical compound, drug | Triton X-100 | Sigma-Aldrich | X100-500ML | |
| Chemical compound, drug | *N*-Methylacetamide, ≥99% | Sigma-Aldrich | M26305-100G | |
| Commercial assay or Kit | V3.0 HCR kit | Molecular Instruments | | |
| Chemical compound, drug | Abberior Mount, Liquid Antifade | Abberior GmbH | MM-2009-2 × 15 ML | |
| Chemical compound, drug | Tissue Tek O.C.T. Compound | Science Service | 62550-12 | |

Axolotl Husbandry d/d Axolotls were maintained in individual aquaria and all animal matings were undertaken in the IMP animal facility. All axolotl handling and surgical procedures were performed in accordance with local ethics committee guidelines. Animal experiments were performed as approved by the Magistrate of Vienna (animal license number GZ: 9418/2017/12).

## Sample harvesting

Prior to amputation, axolotls were anesthetized in 0.03% benzocaine (Sigma) solution until they no longer responded to physical stimuli. For expression profiling axolotl limb buds were harvested at stage 45–46. For expression profiling of blastema samples, forelimbs of axolotl snout-to-tail 3–4 cm were amputated through the middle of the zeugopods and allowed to regenerate for 7 days before harvesting. Samples were fixed in 4% Paraformaldehyde PFA at 4°C overnight (14–17 hr) and then dehydrated progressively in a series of methanol–phosphate-buffered saline (PBS): 25–75%, 50–50%, 75–25%, and 100% methanol, each step on ice for at least 30 min. Samples were stored until usage at −20°C in 100% methanol. Prior to staining, limbs were rehydrated in progressively in 25–75%, 50–50%, 75–25%, and 100–0% PBS–methanol.

## Drug treatments

Axolotls 3–4 cm from snout-to-tail were amputated as described above. Drug treatments were performed for different durations always starting at 5-day postamputation.

For the **12-hr IWR treatment,** 4 axolotls were treated for 12 hr with 25 µM IWR-1-endo (I0161-25MG, Sigma-Aldrich, dissolved in DMSO), in a 30 ml water bath, and 4 axolotls with the corresponding amount of DMSO as a control. All samples from this experiment were stained and imaged together on the same day.

For the **12-hr CHIR treatment,** 4 axolotls were treated for 12 hr with 50 µM CHIR-99021 (4423 Tocris, dissolved in DMSO), in a 30-ml water bath, and 4 axolotls with the corresponding amount of DMSO as a control. All samples from this experiment were stained and imaged together on the same day.

For the **3- and 6-hr CHIR treatment**, 6 axolotls were treated with 50 µM CHIR or a DMSO control for either 3 or 6 hr, in a 15-ml water bath. Limbs were stained and imaged in two rounds.

## Whole-mount HCR in situ hybridization and tissue clearing

HCR probe pairs were designed against unique mRNA sequences common to all isoforms of the candidate gene, with the exception of Amex.*Fgfr2b* for which isoform-specific probes were designed. Unique transcript sequences were identified by BLAST alignment of the sequence against the axolotl transcriptome Amex.T_v47 (*Schloissnig et al., 2021*) and eliminating transcript regions with a homology to nontarget gene transcripts larger than 36 out of 50 nucleotides. Probes for Amex.*Fgf8*, Amex.*Fgf10*, Amex.*Fgfr2b*, and Amex.*Prrx1*, hybridization wash and amplification buffers, as well as HCR hairpins were purchased from Molecular Instruments. The remaining probes were ordered as IDT Opools. For signal amplification, exclusively Alexa-546- and Alexa-647-conjugated hairpins were used in this study.

Limbs were stained using the whole-mount HCR protocol from *Choi et al., 2018* with the following modifications. DAPI (4,6-diamidino-2-phenylindole Sigma, #D9542) was added with the hairpins at the amplification step. After washing in 5× Saline-Sodium Citrate Tween (0.1%) buffer (SSCT), samples were washed for 2 × 30′ in PBS and then cleared overnight incubating at room temperature in Ce3D solution. The Ce3D clearing media was prepared according to published methods (*Li et al., 2017*; *Li et al., 2019*), but without the use of 1-Thioglycerol (*Anderson et al., 2020*). Whole-mount imaging was performed using a Z1 lightsheet microscope equipped with a ×20 detection clearing objective (×20/1.53: Clr Plan-Neofluar ×20/1.0 Corr DIC; nd = 1.53, WD = 6.4 mm) and ×10/0.2 illumination objectives. The detection objective was set to match the RI of the clearing media (1.501) prior to imaging. HCR signal was excited using the 647 and 561 nm excitation wavelength lasers, while 488 nm illumination was used to detect autofluorescence. DAPI was illuminated with the 405 nm laser.

## Section HCR staining

After fixation, dehydration, and rehydration, limbs were incubated overnight at 4°C in 30% sucrose/PBS, transferred to OCT compound (Science Service), frozen on dry ice, and sectioned using a cryostat to a thickness of 12 µm. HCR staining was performed according to the *Choi et al., 2018* protocol for samples on a section. Sections were mounted using Abberior liquid antifade mounting media. Slides were imaged with a Zeiss microscope LSM 980 (inverted Axio Observer with Airyscan) using a ×20/0.8 plan-apochromat air objective and the MPLX SR-4Y mode. Raw images were Airyscan processed and stitched using standard parameters in Zen software.

## Quantification of whole-mount data

All image analysis was performed in Fiji (*Schindelin et al., 2012*). First, images were rotated and cropped in 3D using CLIJ2 (*Haase et al., 2020*), distal to the ulna and radius to exclude the mature stump and isolate the blastema. Subsequently, background levels were calculated for each HCR channel by averaging multiple measurements of a 5 μm × 5 μm region of the epidermis devoid of any HCR signal and subtracted for each HCR channel.

Axolotl limb buds and blastemas contain blood cells that are highly auto fluorescent and can compromise the accuracy of signal quantifications. To address this issue, we acquired an image of the autofluorescence in an empty channel (488-nm illumination). We then segmented the gene expression domains, as well as the autofluorescent signal and subtracted autofluorescent regions from the gene expression domains.

To segment gene expression domains or autofluorescent structures, we first denoised the data using a 3D Gaussian blur (CLIJ2) with a sigma of $x = 6.0$, $y = 6.0$, $z = 1.74$ pixels, corresponding to 3.84 μm in all directions. We then thresholded manually the resulting image, isolating the expression domains in the 647- and 546-nm channel or autofluorescent structures in the 488-nm channel. Different thresholds were used for each channel, experiment, and imaging session, to account for fluctuations in laser power or staining efficacy but the same threshold was applied to all samples of the same experiment. The resulting segmentation masks were applied to the cropped raw data, after background subtraction, for 3D average intensity measurement using the 3D Roi Manager in Fiji (*Ollion et al., 2013*).

To measure *Axin2* expression in the epidermis, the mask of Amex.*Fgf8* expression domain was subtracted from the *Axin2* mask to isolate the epidermal Amex.*Axin2* expression domain (a low threshold for the Amex.*Fgf8* segmentation allows isolation of the entire mesenchymal domain, *Figure 5—figure supplement 3*). The mesenchymal Amex.*Axin2* expression domain was further isolated by subtracting the epidermal mask from the Amex.*Axin2* mask.

The procedure was executed in automatic batch mode using Fiji macro scripting, statistical analysis, and plotting were done using R software and the ggplot2 library (*Wickham, 2016*; *RStudio Team, 2020*; *R Development Core Team, 2021*).

## Statistical analysis

For all experiments performed, the number of replicates was decided empirically, and no samples were excluded. Different samples for each individual whole-mount HCR staining (corresponding to figure panels) in *Figures 2 and 4* were processed, stained, and imaged in the same experiment.

For all pharmacological treatments sibling, animals were randomly allocated to the treatment or control group. For the 3- and 6-hr CHIR treatment, pharmacological treatment was performed once, and limbs were stained and imaged in two rounds (4 limbs per condition per round). Data were normalized only to the DMSO samples imaged in the same round. After normalization, the effect on Amex.*Fgf8* expression for the two different staining rounds resulted not statistically different (Wilcoxon rank sum), so samples were combined for further analysis. For the 12-hr CHIR treatment, pharmacological treatment was performed once, and limbs were stained and imaged together. There are eight data points for Amex.*Fgf8* and four for Amex.*Axin2*, because four of the Amex.*Fgf8* stained samples were costained with a different marker. For the 12-hr IWR treatment, pharmacological treatment was performed once, and limbs were stained and imaged together.

After rotation and cropping, image analysis was performed automatically and with identical parameters for treatment and corresponding control groups. For the evaluation of the effect of pharmacological Wnt signaling perturbation on Amex.*Axin2* and Amex.*Fgf8* expression, the mean HCR intensity fluorescence in the segmented gene expression domains for a single blastema was used as a data point. Samples were grouped according to the treatment condition, and intensity data for each data point were normalized to the average of the DMSO control samples for the corresponding staining and treatment condition. Statistical comparison between sample groups and the corresponding DMSO controls was performed using the nonparametric Wilcoxon rank sum test.

Analogous statistical analysis was performed to compare Amex.*Axin2* HCR signal intensities in epidermis and mesenchyme (*Figure 5E*).

## Prrx1:Cre lineage tracing

Prrx1:Cre-ER;CAGGs:lp-Cherry axolotls were generated and tamoxifen converted during limb bud stages as described previously, to irreversibly label Prrx1 lineage mesenchymal cells (*Gerber et al., 2018*). Samples were processed for sectioning, staining, and imaging as described above.

## Single-cell analysis

Single-cell data were reanalyzed to match the latest axolotl transcriptome annotation (*Schloissnig et al., 2021*). The data were then imported into RStudio as a Seurat object, a data class specifically designed for manipulating scRNA seq data (*Hao et al., 2021*). Dimensionality reduction was performed using the FindClusters() function, with a resolution of 0.05—the resolution determines how many clusters will be generated from the data. To identify which clusters represent mesenchyme and epidermis, we plotted several marker genes of mesenchyme (Amex.*Prrx1*, Amex.*Fgf10*) and epidermis (Amex.*Krt12*, Amex.*Col17a1*, Amex.*Frem2*, Amex.*Epcam*) using the FeaturePlot() function. For *Figures 1 and 3*, clusters were renamed 'mesenchyme' or 'epidermis' if they expressed corresponding marker genes, and all other clusters were made nameless. Markers for the other clusters are shown in *Figure 1—figure supplements 1–2*. All subsequent plots were done using the FeaturePlot() function.

## Figure compilation

All images were processed in Fiji and adjusted in brightness and contrast only for clarity (without gamma adjusting). Figure data plots were generated using R software. Figures were assembled in Adobe Illustrator.

## Acknowledgements

We thank Francisco Falcon for reanalyzing the single-cell datasets to match the latest axolotl transcriptome annotation. We wish to thank Pavel Pasierbek and Alberto Moreno Cencerrado at the IMP Biooptics for their support. We thank Akane Kawaguchi and Leo Otsuki for advice and Magdalena Blaschek, Emina Silic, Tamara Torrecilla Lobos, and Viktoria Szilagyi for axolotl care. We thank Osvaldo Chara and András Aszodi for statistical advice. This work was supported by a Marshall Plan Foundation scholarship to GLG, and ERC AdG 7420346 and FWF I4846 grants to EMT.

## Additional information

### Funding

| Funder | Grant reference number | Author |
|---|---|---|
| Marshallplan-Jubiläumsstiftung | 1062 1304 14 24 2020 | Giacomo L Glotzer |
| H2020 European Research Council | AdG 7420346 | Elly M Tanaka |
| Austrian Science Fund | I4846 | Elly M Tanaka |
| Research Institute of Molecular Pathology | | Pietro Tardivo |

The funders had no role in study design, data collection, and interpretation, or the decision to submit the work for publication.

### Author contributions

Giacomo L Glotzer, Conceptualization, Data curation, Formal analysis, Investigation, Methodology, Software, Validation, Visualization, Writing – original draft, Writing – review and editing; Pietro Tardivo, Conceptualization, Data curation, Formal analysis, Investigation, Methodology, Software, Supervision, Validation, Visualization, Writing – original draft, Writing – review and editing; Elly M

Tanaka, Conceptualization, Formal analysis, Funding acquisition, Investigation, Methodology, Project administration, Resources, Supervision, Validation, Visualization, Writing – original draft, Writing – review and editing

### Author ORCIDs
Giacomo L Glotzer http://orcid.org/0000-0003-2404-6110
Pietro Tardivo http://orcid.org/0000-0001-7878-5272
Elly M Tanaka http://orcid.org/0000-0003-4240-2158

### Ethics
All axolotl handling and surgical procedures were performed in accordance with local ethics committee guidelines. Animal experiments were performed as approved by the Magistrate of Vienna (animal license number GZ: 9418/2017/12).

### Decision letter and Author response
Decision letter https://doi.org/10.7554/eLife.79762.sa1
Author response https://doi.org/10.7554/eLife.79762.sa2

---

## Additional files

### Supplementary files
• Supplementary file 1. HCR probes sequences and transcripts IDs table.

• Transparent reporting form

### Data availability
Figure 5- source data contains the numerical data used to generate the figure Fiji macros used for image analysis have been uploaded to Github: https://github.com/labtanaka/glotzer_fiji_scripts, (copy archived at swh:1:rev:470caf813f9f2a1f13f60a6791fa562db6466660).

The following previously published datasets were used:

| Author(s) | Year | Dataset title | Dataset URL | Database and Identifier |
| --- | --- | --- | --- | --- |
| Lin TY, Gerber T, Taniguchi-Sugiura Y, Murawala P | 2021 | Fibroblast Dedifferentiation as a Determinant of Successful Regeneration | https://www.ncbi.nlm.nih.gov/geo/query/acc.cgi?acc=GSE165901 | NCBI Gene Expression Omnibus, GSE165901 |
| Li H, Wei X, Zhou L | 2021 | blastema 7dpa | https://www.ncbi.nlm.nih.gov/sra/?term=SRX7140465 | NCBI Sequence Read Archive, SRX7140465 |

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
