## [Editor Report]

The authors investigate the upstream regulators of axolotl Fgf8 expression in limb development and regeneration with the aim of identifying the reasons for its unique mesenchymal localization. The results suggest some refractoriness of the axolotl epidermis to activate Wnt signaling although Fgf8 expression in the anterior mesenchyme is still under Wnt control.

---

## [Decision Letter]

[Editors' note: this paper was reviewed by Review Commons.]

---

## [Author Response]

Reviewer #1 (Evidence, reproducibility and clarity (Required)):I have three comments on the data interpretation from the figures as I see them.1. Fgf10 expression.Page 6 says 'Surprisingly, we additionally found Amex.Fgf10 transcripts in a small and distinct cell population located in the basal layer of the distal epidermis. To exclude that this population is of mesenchymal origin we performed HCR in situ hybridization and lineage tracing using Prrx1-Cre transgenic axolotls (Supp. Figure 1).And on page 9 it says 'In our study we also report that Amex.Fgf10 expression is present not only in the axolotl limb mesenchyme but also in a small cell population localized in the basal layer of the limb bud and blastema epidermis.The images in Supplementary figure 1 do not show the purple dots in the basal epidermis, they are on the outer layers of the epidermis (especially supplementary figure 1B).There is a whole paragraph on the hypothesis that epidermal Fgf10 is playing an important role based on something I can't see in the figures.

We confirm that we detect Amex.Fgf10 in a portion of the basal epidermis of the axolotl blastema and limb bud. Basal epidermis expression is visible, albeit weak, in Figure 4B and 4B’ of our updated manuscript. We have added arrows to indicate the basal epidermis expression in this figure to facilitate the reader’s interpretation. To make our imaging data easier to evaluate, we magnified all the panels in Figure 4.

We also confirm that in Supplementary Figure 4 of the current version of the manuscript Amex.Fgf10 is detected in the basal epidermis. To facilitate the interpretation of the figure we enhanced the DAPI channel contrast, making the outer epidermal layers more visible. We also labeled the box outlined in supplementary Figure 4-figure supplement 1A4A, indicating that it corresponds to the region magnified in Figure 4-figure supplement 1B and 1C.

2. Fgfr2 expression.Page 6 says 'In the axolotl, using HCR, we detected Amex.Fgfr2b transcripts through the entire basal epidermis of the blastema (Figure 4B) and limb bud (Figure 4B'), consistent with Amex.Fgf10 signaling to the epidermis as in other vertebrates.The image in Figure 4B shows Fgfr2b (purple) in the apical layers of the epidermis, not the basal.Therefore It is unlikely that Fgfr2b is playing a signaling role between the epidermis and mesenchyme via fgfg10 as it is 'so far away' from the mesenchyme.

We thank the reviewer for pointing out this additional lack of clarity in one of our figures. In the previous version of our figures, the right magenta structures in the blastema outer epidermis corresponded to autofluorescence and did not represent specific HCR signal. We acquired new data for Fgfr2b expression and updated the figure panel (Figure 4C) with a clearer image. We also updated the limb bud Fgfr2b image (Figure 4C’).

3. Wnt3a expression. If Wnt 3a is expressed in the mesenchyme why don't you see it in the single cell data in Figure 3B, B'? I don't see any Wnt3a expression in the mesenchyme in Figure 4, perhaps some dots in the limb bud (B') but certainly not in the blastema mesenchyme (B).Whether these genes are expressed in particular domains or not is important for constructing hypotheses so it should be clear to the reader.

We thank the reviewer for raising this important point. We confirm that we detect Wnt3a transcripts mainly in the basal epidermis of the axolotl limb bud and blastema. We acquired new data for Wnt3a expression and updated the corresponding figure panels (Figure 4D, D’). We agree with the reviewer that the images show expression in the basal epidermis, consistent with the single cell analysis. We therefore removed the sentence regarding expression of Wnt3a in the mesenchyme. We think that basal epidermis is the main site of Wnt3a expression in the axolotl limb, as visualized by both the single-cell analysis and in situ hybridization. This is relevant in consideration of the fact that Wnt3a expression is sufficient to induce ectopic Fgf8 expression in the chick limb epidermis, and Wnt3 expression is necessary for Fgf8 expression in the mouse limb epidermis. The axolotl epidermis lacks Fgf8 expression despite the presence of this putatively upstream ligand. Downstream Wnt signaling activation appears in the axolotl biased toward the mesenchyme, as reflected by the Axin2 co-stain in Figure 4D, 4D’.

Reviewer #1 (Significance (Required)):Valuable data on the relationship between signaling genes during axolotl limb development and regeneration. All our current concepts on this subject are based on the chick and the mouse developing limbs, but these can't regenerate. Do regenerating limbs have different gene relationships during development which could explain why the axolotl can regenerate and other limbs can't.Reviewer #2 (Evidence, reproducibility and clarity (Required)):This is a very interesting study investigating the upstream regulators of axolotl Fgf8 expression in limb development and regeneration with the aim of identifying the reasons for its unique mesenchymal localization. Using scRNAseq and HCR, the authors identify both conserved and divergent gene expression features. Overall, the results suggest some refractoriness of the axolotl epidermis to activate Wnt signaling although Fgf8 expression in the anterior mesenchyme is still under Wnt control.I will mention a few points for consideration by the authors.- One important point is the stage at which the study has been performed (particularly in limb development), regarding the activation and/or maintenance of Fgf8 expression. It is known that Wnt signaling is continuously required for Fgf8-AER expression both in mouse and chick, but in principle the inductor could be present only for the initial activation. The authors may consider discussing the difference between activation and maintenance. Also, I think it would be helpful to show a drawing/picture of a St 52 axolotl hindlimb indicating the corresponding mouse/chick stages.

We thank the reviewer for giving us the opportunity to clarify this point. Our expression and pharmacological perturbation data is acquired for both the limb bud and blastema at stages subsequent to the initiation of Fgf8 expression, and therefore addresses the maintenance aspect of Fgf8 expression. We have clarified this in the Discussion section of the manuscript:

“A positive effect of CHIR treatment on Amex.*Fgf8* expression was present after 3h of treatment but disappeared when the perturbation was performed for longer durations, possibly due to indirect effects or negative feedbacks triggered by the ubiquitously high Wnt signaling environment. The positive upstream regulatory role of Wnt signaling is similar to that seen in chicken and mammals yet occurring in a different tissue compartment (Barrow, Thomas et al. 2003, Soshnikova, Zechner et al. 2003). Our experiments have been performed at stages of limb development and regeneration subsequent to the initiation of Amex.*Fgf8* expression and therefore indicate a function of canonical Wnt signaling at least in maintaining Amex.*Fgf8* expression. In mouse and chick, canonical Wnt signaling is required for both the initiation and maintenance of endogenous Fgf8 expression and, when activated ectopically, can induce ectopic epidermal Fgf8 expression (Kengaku, Capdevila et al. 1998, Galceran, Farinas et al. 1999, Barrow, Thomas et al. 2003, Soshnikova, Zechner et al. 2003, Kawakami, Esteban et al. 2004). Interestingly, in the axolotl, canonical Wnt activation was not sufficient to induce ectopic Amex.*Fgf8* expression in the ectoderm.”

We have also mentioned explicitly in the main text (as quoted below) that HCR in situ hybridizations were performed on stage 44-45 axolotl forelimbs. We think forelimbs at this stage are analogous to the stage 52 hindlimb buds that were used for the generation of the single cell dataset analyzed. We decided to perform our experiments in the forelimb to be consistent with the blastema single cell dataset analyzed and with the large corpus of available limb literature that focuses on the forelimb.

“To this end, we first analyzed published scRNA seq datasets for the axolotl stage 52 axolotl hindlimb bud (Lin, Gerber et al. 2021) and 7 days-post-amputation axolotl forelimb blastema (Li, Wei et al. 2021). Both single cell datasets were reanalyzed to match the latest axolotl transcriptome annotation (Schloissnig, Kawaguchi et al. 2021) and re-clustered, using known markers for cluster assignment (Figure 1-figure supplements 1 and 2). We then used HCR in situ hybridization in tissue sections and whole-mount cleared limb preparations of forelimb buds and blastemas of comparable stages (stage 45-46 forelimb bud, 7 days post amputation forelimb blastema) to validate the expression domains and acquire spatial information.”

Although we are confident that the stages analyzed represent the maintenance phase of Fgf8 expression, we are unable to directly compare stages with that of mouse and chick, especially given that the genes we examine show different expression patterns.

- Is the number of Sp6 expressing cells in the mesenchyme cluster negligible? Is Sp6 expression in the mesenchyme detected by HCR?

We thank the reviewer for raising this question. We assessed Sp6 expression in the limb bud and blastema using HCR and included the images in Figure 2 and in a new Figure 2-figure supplement 1. We refer to this new data in our manuscript as follows:

“We performed whole mount HCR staining in regenerating axolotl limb blastemas and limb buds (Figure 2B, 2B’) and confirmed that Amex.*Sp6* is expressed in the epidermis of the axolotl limb bud and regenerating blastema. Interestingly, we detected Amex.*Sp6* also in the mesenchyme of the axolotl limb bud, in a domain proximal to Amex.*Fg8* expression (Figure 2-figure supplement 1). This proximal mesenchymal expression of Amex.*Sp6* was absent in the blastemas that we analyzed and is unlikely to regulate the more distal Amex.*Fg8* expression in the axolotl limb bud.”

Would it be possible to have an idea of the nameless cluster continuation of the epidermal cluster that contains many Sp8 positive cells.

The additional cluster that contains Sp8 positive cells corresponds to epidermal small secretory cells. We now provide two new supplementary figures (Figure 1-figure supplements 1 and 2), where we name the different clusters for the single cell datasets and show UMAPs for some of the markers used to assign the clusters to specific cell populations.

Is the small population of Fgf10 expressing cells in the epithelial layer only found in the blastema or also in limb buds? Can this impact the use of Fgf10 as marker to identify the mesenchymal cluster?

The small population of Fgf10-expressing epithelial cells is also present in limb buds. We indicated epidermal Fgf10 expression in figure 4B and 4B’ of the updated manuscript with an arrow to facilitate the interpretation from the reader. We think that indeed Fgf10 is not a reliable mesenchymal marker. We identified the mesenchymal cluster in our analysis using different markers (Prrx1) as is now visible in the new supplementary figures Figure 1-figure supplements 1 and 2.

Prrx1 is considered a mesenchymal marker however its expression is preferentially seen in some regions of the mesenchyme (Figure 4A', C'). Does Prrx1 expression show dV or other bias?

While we detect Prrx1 transcripts in all the blastema mesenchyme, we agree with the reviewer’s observation that expression of Prrx1 is spatially biased. The bias is present along the anterior-posterior axis in the images from our study. Expression of Prrx1 in present in the entire blastema mesenchyme, but stronger anteriorly.

The results with CHIR are very clear after 3h treatment but fading afterward disregarding Wnt signalling remaining upregulated according to axin. Have the authors considered a possible explanation?

We agree with the reviewer that the effect of CIHIR treatment on Fgf8 expression changes after the 3h treatment. We think that this could be due to indirect effects or negative feedback triggered by prolonged exposure to an induced high Wnt signaling environment. We reported this possibility in the manuscript discussion as follows:

“A positive effect of CHIR treatment on Amex.*Fgf8* expression was present after 3h of treatment but disappeared when the perturbation was performed for longer durations, possibly due to indirect effects or negative feedbacks triggered by the ubiquitously high Wnt signaling environment.”

Please, indicate the AP or DV axis in the limb bud HCR images.

We have not indicated the AP or DV axis in the limb buds, as without the ulna and radius as a reference, it is difficult to reliably distinguish between the axes when specific markers are not stained.

Please, indicate the number of samples analysed for each gene expression.

We indicated the number of samples analyzed in the figure legends.

Reviewer #2 (Significance (Required)):I think this study addresses a topic of interest to a broad audience, is clearly presented and opens new avenues of research towards deciphering the uniqueness of the mesenchymal Fgf8 expression in the axolotl.Reviewer #3 (Evidence, reproducibility and clarity (Required)):Summary:The manuscript by Glotzer et al. tries to address if Wnt signaling is responsible for the mesenchymal expression of FGF-8 in axolotl limbs buds and limb blastemas. Axolotls are different than mice and chicks in that FGF-8 is not expressed in epidermal cells at all. The authors used already available scRNAseq dataset to look at the expression of genes known to play a role in mediating/controlling the expression of FGF-8 in the AER of mice and chicks. The conclusion is that some unidentified gene is likely responsible for the fact that FGF-8 is restricted to mesenchymal cells in axolotls and or the epidermal cells in mice and chicks.Major comments:The key conclusions are that most of the genes expressed in mice and chick developing limbs can be observed during axolotl limb development and regeneration. Another conclusion is that most genes involved in FGF-8 regulation are also expressed in axolotls similar to what is observed in mice and chicks even though FGF-8 is restricted to mesenchymal cells.The data and the methods are presented in such a way that they can be reproduced and references are provided in an appropriate manner.The experiments are adequately replicated and statistical analyses are adequate.Minor comments:Specific experimental issues that are easily addressable.In Figure 2 B and C the stages of the blastema does not seem to be the same. Panel C looks as if it might be a late bud blastema and B an early bud? Instead of a diagram in A and A' an actual picture of a representative blastema and developing limb might be better.

We agree with the reviewer that the Sp8 images in Figure 2 looked different to the other panels, likely because the images in panel B were from tissue sections while the images in the other panels were single planes from whole mount data. We have generated new whole mount data for Sp8 expression and updated the corresponding panels.

We would prefer to keep the diagrams in A and A’, as they allow us to schematically depict the epidermal and mesenchymal compartments.

Figure 4 D' the expression of Rspo2 is not 100% convincing. It would be good to include in the supplementary data the other n for the expression of this gene in axolotl developing limbs.

We have generated additional whole mount data for Rspo2 expression and updated the panels in figure 4 for both the blastema and limb bud (Figure 4 E, E’). We also included another limb bud image in the supplementary data, displaying that a DV bias in Rspo2 expression is present also in the developing limb.

Supp Figure 1: the expression of FGF-10 does not seem to correspond to cells. In panels B and C it looks like blotches that are not specific.

We have enhanced the DAPI contrast in this image. We think that now it is more clearly visible that the HCR signal is in close proximity to the nuclei. HCR signal corresponds to quasi-single molecules of RNA, and preferentially localizes to the cytoplasm and not to nuclei, as expected.

On page 8 the last sentence of the section just before Discussion, it should be Figure 5E and not 3E.

We thank the reviewer for pointing out this mistake that we corrected it in the current version of the manuscript.

Reviewer #3 (Significance (Required)):Describe the nature and significance of the advance (e.g. conceptual, technical, clinical) for the field. Axolotls are unique amongst tetrapod vertebrates in many ways: they can regenerate their limbs and they don't have an AER during limb development or limb regeneration. Interestingly, almost all the genes that are expressed in animals with AER during limb development are also expressed during axolotl limb development. FGF-8 is an important AER marker in mice and chicks. In axolotls FGF-8 is also expressed during both limb development and regeneration but it is never observed in the epidermis. Axo.FGF-8 is exclusively in mesenchymal cells. The authors tried to understand how can this be and what drives the expression of FGF-8 in mesenchymal cells in axolotls as oppose to epidermal cells.The authors demonstrated that multiple genes known to drive the expression of FGF-8 in mice and chicks are also present in axolotls. Some of these genes are in the epidermis (SP6 and SP8) in axolotls although FGF-8 is in the mesenchyme. The true mechanism driving FGF-8 expression in the mesenchyme has not been elucidated in this manuscript but significant new data is presented that helps understand better the similarities and differences between axolotls and other tetrapods.Place the work in the context of the existing literature (provide references, where appropriate). This is one of the first paper that tries to unravel the molecular mechanism behind the expression of mesenchymal FGF-8 expression in axolotls. It has been known for almost 20 years that axolotl FGF-8 is in the mesenchyme. Understanding why it is in the mesenchyme could be important for evolutionary reasons. It could also shed light on other aspects unique to salamanders such as limb regeneration. It is possible that the expression of FGF-8 in the mesenchyme is a crucial aspect of limb regeneration in addition to positional reasons?State what audience might be interested in and influenced by the reported findings.Every scientist with an interest in limb development/regeneration, evolutionary biology of patterning and limb development will be interested in this manuscript.- Define your field of expertise with a few keywords to help the authors contextualize your point of view. Indicate if there are any parts of the paper that you do not have sufficient expertise to evaluate.Limb regeneration, cellular signaling, pattern formation.